# Two *Arabidopsis* Splicing Factors, U2AF65a and U2AF65b, Differentially Control Flowering Time by Modulating the Expression or Alternative Splicing of a Subset of *FLC* Upstream Regulators

**DOI:** 10.3390/plants12081655

**Published:** 2023-04-14

**Authors:** Hee Tae Lee, Hyo-Young Park, Keh Chien Lee, Jeong Hwan Lee, Jeong-Kook Kim

**Affiliations:** 1Division of Life Sciences, Korea University, 145 Anam-ro, Seongbuk-gu, Seoul 02841, Republic of Korea; 2Division of Life Sciences, Jeonbuk National University, 567 Baekje-daero, Deokjin-gu, Jeonju 54896, Jeollabuk-do, Republic of Korea

**Keywords:** splicing factors, flowering time, RNA-Seq, alternative splicing, shoot apical meristem

## Abstract

We investigated the transcriptomic changes in the shoot apices during floral transition in *Arabidopsis* mutants of two closely related splicing factors: AtU2AF65a (*atu2af65a*) and AtU2AF65b (*atu2af65b*). The *atu2af65a* mutants exhibited delayed flowering, while the *atu2af65b* mutants showed accelerated flowering. The underlying gene regulatory mechanism of these phenotypes was unclear. We performed RNA-seq analysis using shoot apices instead of whole seedlings and found that the *atu2af65a* mutants had more differentially expressed genes than the *atu2af65b* mutants when they were compared to wild type. The only flowering time gene that was significantly up- or down-regulated by more than two-fold in the mutants were *FLOWERING LOCUS C* (*FLC*), a major floral repressor. We also examined the expression and alternative splicing (AS) patterns of several *FLC* upstream regulators, such as *COOLAIR*, EDM2, FRIGIDA, and PP2A-b’ɤ, and found that those of *COOLAIR, EDM2*, and *PP2A-b’ɤ* were altered in the mutants. Furthermore, we demonstrated that *AtU2AF65a* and *AtU2AF65b* genes partially influenced *FLC* expression by analyzing these mutants in the *flc-3* mutant background. Our findings indicate that AtU2AF65a and AtU2AF65b splicing factors modulate *FLC* expression by affecting the expression or AS patterns of a subset of *FLC* upstream regulators in the shoot apex, leading to different flowering phenotypes.

## 1. Introduction

The shoot apical meristem (SAM) is formed together with the cotyledons during embryogenesis [1,2,3]. The SAM also undergoes a developmental switch from vegetative to reproductive phase, in which it produces floral meristems (FMs) that give rise to flowers [4]. The main pathways regulating the transition to flowering in *Arabidopsis thaliana* include the photoperiodic pathway, the autonomous pathway, the gibberellin pathway, the aging pathway, the ambient temperature pathway, and the vernalization pathway. The floral signals converge on the flowering integrators *FLOWERING LOCUS T* (*FT*) and *SUPPRESSOR OF OVEREXPRESSION OF CONSTANS1* (*SOC1*) forming a flowering network that sequentially activates the floral meristem identity genes [4,5]. *SOC1* is primarily involved in the upregulation of *AGAMOUS-LIKE 24* (*AGL24*) in SAM during the transition to flowering, and *AGL24* is also involved in the regulation of *SOC1* [6,7]. *SOC1* is also suppressed by *FLOWERING LOCUS C* (*FLC*), a flowering repressor [7]. Alternative splicing (AS) is a common and important mechanism of gene regulation in eukaryotes at posttranscriptional level such as mRNA stability, translation efficiency, and interactions with microRNAs [8]. It can also modulate gene expression in response to various developmental and environmental cues. Among the different types of AS events, intron retention (IR) is the most frequent and distinctive in *Arabidopsis*, accounting for more than 40% of all AS events [9,10,11]. AS is regulated by a complex network of splicing factors that bind to specific cis-elements on pre-mRNAs and interact with the spliceosome machinery.

Among several spliceosome members, the U2 small nuclear ribonucleoprotein (snRNP) auxiliary factor is a group of non-snRNPs involved in early splicing, including the large subunit U2 snRNP auxiliary factor 65kDa (U2AF65) and the small subunit U2 snRNP auxiliary factor 35kDa (U2AF35). In *Arabidopsis*, AtU2AF65a and AtU2AF65b and AtU2AF35a and AtU2AF35b are known as U2AF65 and U2AF35 counterparts, respectively [12,13,14]. In the E-complex assembly stage, U2AF65 interacts with U2AF35 after binding to the 3′ splice site, helping to recognize the branch site (BS) of splicing factor 1 (SF1), a branch point binding protein (BBP) [15,16,17]. Recognition of polypyrimidine tract (PPT) by U2AF65 can be promoted by serine-arginine (SR) protein [15,16].

It has been reported that defects in splicing factors cause abnormal flowering phenotypes in *Arabidopsis* [18]. For example, *atu2af35* mutants show a late flowering phenotype [14], whereas *atsf1* mutants show an early flowering phenotype [13]. In addition, *atu2af65a* and *atu2af65b* mutants exhibit late and early flowering phenotypes, respectively [12,19,20].

There is no clear difference in phenotype could be identified except for flowering time between the *atu2af65a* and *atu2af65b* mutants [12]. This is presumably due to functional redundancy between the two proteins, but these mutants also have opposite flowering time phenotypes. This characteristic can be found in members of various plant gene families. For instance, two isoforms of *Arabidopsis* SR45 proteins play distinct roles in floral and root development [21].

Previously, we found that *FLOWERING LOCUS C* (*FLC*) is a major determinant of the flowering time difference between the *atu2af65a* and *atu2af65b* mutants [12]. The *FLC* transcript levels are increased in the *atu2af65*a mutants but decreased in the *atu2af65b* mutants. However, we do not yet know how *FLC* transcript levels are differentially regulated in these two mutants. In this study, we examined the transcriptomes of these two mutants using RNA sequencing (RNA-seq) to compare their global expression patterns. This may lead to a better understanding of the flowering time differences between these two mutants. The AtU2AF65a and AtU2AF65b genes are expressed at higher levels in some parts of the plant, including the shoot apex, than in other parts [12,20]. Flowering time regulation may occur in two separate compartments of plants, the leaf and SAM. For example, photoperiod signals are integrated into *FT* gene expression in leaves and FT proteins are transported into the SAM to participate in gene regulation. In our previous study, we measured the *FLC* transcript levels using whole seedlings [12]. In this study, we examined shoot apices isolated from young seedlings rather than the whole seedlings for transcriptome analysis because *FLC* expression is much higher in shoot apex than in leaf [22].

Based on RNA-seq analysis of the shoot apices of wild-type (Col-0), *atu2af65a* and *atu2af65b* plants, this study provides evidence for how *Arabidopsis* AtU2AF65a and AtU2AF65b splicing factors play opposing roles in regulating flowering. Furthermore, pre-mRNA splicing defects were analyzed in a subset of genes, and their AS defects were identified using the replicate-multivariate analysis of transcript splicing (rMAT) statistical program [23].

## 2. Results

### 2.1. RNA Sequencing Analysis in the Shoot Apex-Enriched Samples of atu2af65a and atu2af65b Mutants

We previously revealed that two different AtU2AF65 proteins differentially regulate flowering time by modulating the expression of *FLC* [12]. To investigate the global-scale changes in expression levels affected by *atu2af65a* or *atu2af65b* mutation, we performed the RNA sequencing (RNA-seq) analysis in shoot apex (SA)-enriched samples of wild-type (Col-0), two *atu2af65a* and one *atu2af65b* mutant plants. These mutants were chosen based on our previous data for stronger phenotypes [12]. The *atu2af65b* mutants are the same mutant plants used in other laboratory for its transcriptome analysis [20]. We included it for comparison. Although the shoot apices were carefully isolated from 7-d-old seedlings, it is likely that the samples contained a small fraction of emerging leaves and primordia (Appendix A). Total RNA extracted from each sample was used for library construction and sequenced using Illumina protocols. After the removal of low-quality reads, approximately 38, 42, 44, and 45 million uniquely mapped reads were retained for further analysis from wild-type (Col-0), *atu2af65a-4*, *atu2af65a-3*, and *atu2af65b-1* plants, respectively (Appendix A). A heat map was constructed to identify significantly altered genes in the shoot apices of wild-type (Col-0), *atu2af65a-4*, *atu2af65a-3*, and *atu2af65b-1* plants (FDR < 0.05) (Figure 1a and Appendix A). The overall transcriptome profiles of *atu2af65* mutants showed distinct expression patterns compared to those of wild-type (Col-0) plants. Samples of two different T-DNA mutant alleles for *AtU2AF65a* gene were used for RNA-seq analysis, and 2186 genes showing similar expression patterns between the *atu2af65a-4* and *atu2af65a-3* mutants were selected and used as representatives of the genes affected by *atu2af65a* mutation for further analysis (Appendix A). The expression of 2186 genes or 655 genes was affected by either *atu2af65a* or *atu2af65b* mutation, respectively, and the expression of 455 genes was affected by both *atu2af65a* and *atu2af65b* mutations (*q*-value < 0.05) although only 172 and 86 genes were affected by more than two-fold expression level changes in the *atu2af65a* and *atu2af65b* mutants, respectively (Figure 1b).

A total of 1062 and 1119 genes were up- and down-regulated by the *atu2af65a* mutation, respectively, whereas 223 and 427 genes were up- and down-regulated by the *atu2af65b* mutation, respectively (Figure 1b). These results showed that a similar number of genes were up- or downregulated in the *atu2af65a* mutants, whereas the number of downregulated genes in the *atu2af65b* mutants was more than two times that of the upregulated genes. Furthermore, compared to the *atu2af65a* mutation, the *atu2af65b* mutation affected a relatively smaller number of genes, especially the upregulated genes. A total of 152 upregulated and 298 downregulated genes were affected in both *atu2af65a* and *atu2af65b* mutants (Figure 1b). In other words, about 70% of genes affected by the *atu2af65b* mutation were also affected by the *atu2af65a* mutation. However, only about 14% upregulated genes and 27% downregulated genes affected by the *atu2af65b* mutation were also affected by *atu2af65a* mutation. These results showed that more genes appeared to be affected specifically by *atu2af65a* mutation, suggesting that the AtU2AF65a protein may have more specific target pre-mRNAs for its splicing function than the AtU2AF65b protein. 

Only five genes, including *FLC* (*At5g10140*), were identified in the genes that were reversely regulated by either *atu2af65a* or *atu2af65b* mutation (Figure 1c,d and Appendix A). Four genes, except *FLC* were downregulated by *atu2af65a* mutation, but upregulated by *atu2af65b* mutation (Figure 1c). Two genes, *ncRNA* (*At4g04223*) and *FLC* showed more than two-fold change in expression (Figure 1d and Appendix A). These *FLC* expression patterns in the shoot apices of the *atu2af65a* and *atu2af65b* mutants were consistent with previous data generated from whole plants [8,16]. However, the function of other two genes are not known yet, but *SZF1* (*At3g55980*) gene is known to be involved in salt stress resistance [24].

The gene ontology (GO) analysis indicated that several categories of genes were affected in the mutants. In particular, the photosynthesis process related category was significantly affected in *atu2af65a* mutants, whereas the hormonal response related process category was affected in *atu2af65b* mutants (Appendix A).

### 2.2. Downregulation of SOC1 and AGL24 Expression in the Shoot Apices of the atu2af65a Mutants Is Due to Upregulation of FLC Expression 

Because *FLC* was the only flowering time gene showing a clear reverse regulation pattern in *atu2af65a* and *atu2af65b* mutants (Figure 1c,d and Appendix A), we examined 306 genes known for flowering time control in the Flowering Interactive Database (FLOR-ID, http://www.phytosystems.ulg.ac.be/florid accessed on 10 April 2023) to identify the genes responsible for opposite *FLC* expression in these two mutants (Figure 2a). A total of 33 genes were found to be significantly misregulated, of which 12 and 20 were involved in promoting and suppressing flowering, respectively (Figure 2a, Appendix A). We could not find the interaction of these genes accountable for the flowering phenotype of both mutants, except in a few cases. For example, the downregulation of the *SOC1* and *AGL24* genes could be due to the upregulation of *FLC* in the *atu2af65a* mutants (Figure 2b). However, concomitant upregulation of *SOC1* and *AGL24* was not observed in the *atu2af65b* mutants (Figure 2b), although *FLC* expression was decreased (Figure 1c,d and Appendix A). We also confirmed these expression patterns in the leaves (L) and shoot apices (SA) using semiquantitative RT-PCR analysis (Figure 2c). Downregulation of *SOC1* and *AGL24* was clearly detected in the shoot apices of the *atu2af65a* mutants. However, the upregulation of these two genes was not observed in the *atu2af65b* mutants, suggesting that the downregulation of *FLC* may differently regulate its downstream genes in the *atu2af65b* mutants. We found that other flowering time genes, such as *SHORT VEGETATIVE PHASE* (*SVP*), *FLOWERING LOCUS M* (*FLM*), and *FLOWERING LOCUS D* (*FD*), were not affected by either *atu2af65a* or *atu2af65b* mutation (Appendix A).

Since AtU2AF65 proteins are splicing factors, we also examined pre-mRNA splicing defects at the chromatin-bound (CB)-RNA level in the nuclear RNA fraction, which is known to detect co-transcriptional splicing defects [26]. We established the isolation method of total RNA first, and then isolated the cytoplasmic and nuclear RNA fractions from 7-d-old wild-type (Col-0) whole seedlings. The 28S and 18S rRNA bands were confirmed to be much less enriched in nuclear RNA but unspliced *FLC* RNA and the highest enrichment of *miR167a* were found in this fraction (Appendix A). To investigate the expression patterns of the pre-mRNA levels of both *SOC1* and *AGL24* genes, total RNA and cytoplasmic and nuclear RNA fractions from the shoot apices of 7-d-old wild-type (Col-0), *atu2af65a-4*, *atu2af65a-3*, and *atu2af65b-1* plants were isolated. However, we could not detect significant changes in the pre-mRNA levels of both *SOC1* and *AGL24* genes in these two mutants using semi-quantitative RT-PCR and RT-qPCR analyses (Figure 2d and Appendix A), suggesting that the misregulation of these two genes in the *atu2af65a* mutants is probably due to the upregulation of their direct upstream regulator, *FLC*, and not splicing defects of *SOC1* and *AGL24* pre-mRNAs directly by AtU2AF65a. The elevated expression of *FLC* in whole seedlings of the *atu2af65a* mutants is mostly due to the upregulation of other *FLC*-specific regulators such as the FRIGIDA (FRI) complex or autonomous genes [12]. However, the changes in their expression were not detectable in the shoot apices of the *atu2af65a* mutants (Appendix A), indicating that the *FLC* expression may be regulated differently in the shoot apex regions.

We have previously shown that the flowering phenotypes of *atu2af65a* and *atu2af65b* mutants were maintained at lower growth temperatures [8]. In this study, we have constructed the double mutants with *flc-3* mutants to provide genetic evidence that the flowering phenotypes of these two mutants are due to change in their *FLC* expression. Genetic analysis of the *atu2af65a* or *atu2af65b* mutants with *flc-3* mutants also indicated that *flc* mutation suppressed partially the flowering phenotype of *atu2af65a* or *atu2af65b* mutants at 10, 16, and 23 °C (Figure 3), suggesting that *FLC* plays a role in determining the flowering phenotype of these two mutants even at different temperatures.

### 2.3. COOLAIR Is Involved in Misregulation of FLC Expression in the Shoot Apices of the atu2af65a Mutants 

Since we could not find significant changes in the expression of the well-known *FLC* upstream regulators, such as FRIGIDA (FRI) complex genes in the shoot apices of the *atu2af65a* mutants, other genes were examined as possible causes of *FLC* expression in the mutants. *COOLAIR* long non-coding RNAs (lncRNAs), upstream regulators of *FLC* are antisense transcripts involved in *FLC* chromatin silencing and transcriptional repression [27,28]. *COOLAIR* RNAs are classified into *Class I* (proximal isoforms of *COOLAIR*) and *Class II* (distal isoforms of *COOLAIR*) according to the position where polyadenylation occurs by 3′-end processing factors [27,29,30]. Each Class of *COOLAIR* consists of alternatively spliced (AS) transcripts (Figure 4a). The expression patterns of *COOLAIR Class I* and *II* in *atu2af65a* and *atu2af65b* mutants were examined by using semi-quantitative RT-PCR and RT-qPCR analyses. A decrease in *COOLAIR Class I* (ii) and an increase in *COOLAIR Class II* (ii)’ were found in *atu2af65a-4* and *atu2af65a-3* mutants (Figure 4b), suggesting that *FLC* level changes in the *atu2af65a* mutants were concomitant with *COOLAIR* level changes. Furthermore, the ratio of *COOLAIR Class I* vs. *Class II* in *atu2af65a*, but not in *atu2af65b*, was approximately 0.26, indicating reduced usage of the *COOLAIR* proximal poly(A) site and increased use of the distal site (Figure 4c). However, there were no significant changes in *COOLAIR* transcripts (*Class I* and *II*) in *atu2af65b-1* mutants (Figure 4b,c). Furthermore, overexpression of AtU2AF65a or AtU2AF65b did not affect *COOLAIR* transcript levels (Figure 4b). We also examined the expression of *COOLAIR* transcripts (*Class I* and *II*) in *atu2af65a-4 flc-3* and *atu2af65b-1 flc-3* double mutants. The expression patterns of decrease in *COOLAIR Class I* and increase in *COOLAIR Class II* were observed in the *atu2af65a-4 flc-3* double mutants, but not in *the flc-3* single and *atu2af65b-1 flc-3* double mutants (Figure 4d), suggesting that AtU2AF65a may affect the regulation of *COOLAIR* expression.

Other splicing factors, such as PRP8 and NSRa, are known to be involved in *COOLAIR*-dependent *FLC* expression [30,31,32]. To examine the possible direct effect of PRP8 and NSRa proteins contributing to the expression pattern changes of *COOLAIR* transcripts, we examined their expression in *atu2af65a* and *atu2af65b* mutants. Only *PRP8* expression increased slightly in *atu2af65a* and *atu2af65b* mutants (Appendix A), which did not match the decrease in *COOLAIR Class I*. No differences were observed in the expression of *FPA* and *NSRa* (Appendix A). These results suggest that the change in the pattern of *COOLAIR* expression in *atu2af65a* mutants is due to the direct effect of AtU2AF65a protein deficiency.

### 2.4. EDM2 and PP2A-b’ɤ Genes Are Involved in Change in FLC Expression in the atu2af65a or atu2af65b Mutants

Our RNA-seq data revealed that only a limited number of abnormal splicing events were altered in either *atu2af65a* or *atu2af65b* mutants (Appendix A). This suggests that most of the co-transcriptional splicing defects in these mutants are probably undetected because of degradation by RNA surveillance systems, such as nonsense-mediated decay [33]. Because of the changes detected in the expression of the *COOLAIR* transcripts in the *atu2af65a* mutants (Figure 4), the expression of *EDM2* (*At5g55390*) and *PP2A-b’ɤ* (*At4g15415*), which are known to be upstream negative regulators of *FLC* expression, were also examined [34,35]. RT-qPCR analysis revealed a decrease in expression of *EDM2* and *PP2A-b’ɤ* in the *atu2af65a* mutants, suggesting that the decreased expression of two genes could explain up-regulation of *FLC* expression in the *atu2af65a* mutants (Figure 5a). These changes in expression were also observed in *atu2af65a-4 flc-3* double mutants (Figure 5b). Furthermore, RNA-seq data revealed that the relative ratio of AS transcripts for each gene was differentially changed in both mutants (Figure 5c). In the *atu2af65a* mutants, only *PP2A-b’ɤ* gene showed significant change in AS, whereas both genes showed significant change in the *atu2af65b* mutants (Figure 5c). AS occurs in the 5′UTR regions of both *EDM2* and *PP2A-b’ɤ* pre-mRNAs, resulting in identical proteins but possibly different translation efficiencies. The AS patterns of these genes were affected by *atu2af65a* or *atu2af65b* mutation. In *atu2af65a* mutant, *EDM2* AS is unchanged but *PP2A-b’ɤ* AS is altered, suggesting that *PP2A-b’ɤ* AS may influence *FLC* expression. In *atu2af65b* mutant, both *EDM2* and *PP2A-b’ɤ* AS were changed, but *EDM2* AS change was more significant than *PP2A-b’ɤ* AS change. However, the total *EDM2* transcript level was not affected by *atu2af65b* mutation, implying that its AS change may have a minor effect on *FLC* expression (Appendix A). These results suggest that differential expression changes in AS transcripts of two genes in *atu2af65a* and *atu2af65b* mutants may affect *FLC* expression, leading to opposite flowering times in these mutants.

### 2.5. Global-Scale Changes in Splicing Events in the atu2af65a and atu2af65b Mutants

In addition to the pre-mRNA splicing events of flowering time genes, the global profiling of splicing events in shoot apices of wild-type (Col-0), *atu2af65a-4*, *atu2af65a-3*, and *atu2af65b-1* plants were also examined. Altered splicing events detected by the statistical program rMAT in RNA-seq data were examined. Compared to Col-0, we found changes in all five types of AS events in *atu2af65* mutants (Figure 6a,b): (1) skipped exon (SE), (2) alternative 3′ splice site (A3SS), (3) retained intron (RI), (4) alternative 5′ splice site (A5SS), and (5) mutually exclusive exons (MXE). The number of altered AS events in the mutants ranged from 0 to 234 depending on the AS type (Figure 6a). The number of altered AS events in *atu2af65a* mutants was higher than that in *atu2af65b-1* mutant. For example, there were 205 altered splicing events for the SE type in the *atu2af65a-4* mutant but in *atu2af65b-1* mutant there were only 131 events. The total number of altered splicing events was 676 in the *atu2af65a-4* mutant and 498 in *atu2af65b-1* mutant. These results indicate that *atu2af65a* mutation may affect global-scale splicing events to a greater extent than *atu2af65b* mutation. However, both *atu2af65a* and *atu2af65b* mutations showed similar patterns for the proportion of all altered AS types, indicating that the two mutations have similar impacts on the distribution of altered AS types (Figure 6b).

The differentially expressed genes (DEG, *p*-value < 0.05) of AS genes affected by either the *atu2af65a-4* or *atu2af65b-1* mutation were compared. The results of rMAT analysis showed that 37 and 15 genes contained AS transcripts that were significantly affected by *atu2af65a* or *atu2af65b* mutations, respectively (Appendix A).

To confirm the alterations in splicing events shown in Appendix A, we selected five genes (*At1g09140*, *At1g73470*, *At4g00560*, *At5g37850*, and *At4g36690*) and examined their expression levels and splicing efficiencies in *atu2af65a-4* and *atu2af65b-1* mutants using semi-quantitative RT-PCR and RT-qPCR analyses. In *atu2af65a-4* mutants, *At1g09140* and *At1g73470* showed misspliced types of A3SS and RI, respectively (Appendix A). We found that the altered A3SS form (*At1g09140*.2) decreased, and the intron retention (RI) form (*At1g73470.2*) increased in only *atu2af65a-4* mutants (Figure 6c). In *atu2af65b-1* mutants, *At4g00560*, *At5g37850*, and *At4g36690* showed one or two misspliced AS types (Appendix A). We found that the SE form (*At4g00560.2*) and altered A5SS form (*At5g37850.2*) increased, and the altered A3SS form (*At4g36690.3*) decreased in only *atu2af65b-1* mutants (Figure 6d). These results indicate that *atu2af65a* and *atu2af65b* mutations lead to different splicing events at the global scale.

### 2.6. Alternative Splicing of AtU2AF65a Pre-mRNA could Be Altered Similarly by Mutations in AtU2AF65b and AtSF1 Genes as Well as Lower Ambient Growth Temperature

Global-scale splicing pattern analysis showed that the functional *AtU2AF65a* form (*At4g36690.1*) was increased in the shoot apices of *atu2af65b* mutants. Therefore, the expression of two *AtU2AF65a* alternatively spliced (AS) forms (*At4g36690.1* and *At4g36690.3*) in whole seedlings of wild-type (Col-0), *atu2af6a-4*, *atu2af65b-1*, and *atsf1-2* plants were also examined. We found that the functional *AtU2AF65a* form (*At4g36690.1*) increased in the *atu2af65b-1* and *atsf1-2* mutants, whereas the other spliced form (*At4g36690.3*) decreased (Figure 7a). This suggests that functional deficiency of AtU2AF65b and AtSF1 proteins may cause an increase in AtU2AF65a protein levels to compensate for their loss in pre-mRNA splicing machinery. Because differences in ambient temperature affect the splicing patterns of splicing factors [19], the expression of the three *AtU2AF65a* spliced forms (*At4g36690.1*, *At4g36690.2*, and *At4g36690.3*) and one *AtU2AF65b* transcript (*At1g60900.1*) in whole seedlings of wild-type (Col-0), and *atu2af6a-4*, *atu2af65b-1*, and *atsf1-2* mutants grown at different ambient temperatures (23 °C and 16 °C) was investigated. A lower temperature (16 °C) induced an increase in the functional *AtU2AF65a* form (*At4g36690.1*) and a decrease in the spliced form (*At4g36690.3*) (Figure 7b). It also decreased *AtU2AF65b* transcript levels in wild-type (Col-0) (Figure 7b). That is, the lower temperature condition mimicked the effect of the *atu2af65b* mutation. Similar changes were detected in the relative ratio of the two *AtU2AF65a* spliced forms (*At4g36690.1* and *At4g36690.3*) in the *atu2af65b-1* and *atsf1-2* mutants at lower temperatures (Figure 7b). However, no *AtU2AF65b* splicing alteration (*At1g60900.1*) was observed in the *atu2af65a-4* and *atsf1-2* mutants (Figure 7c). These results suggest a shared compensation mechanism under these conditions. 

## 3. Discussion

There are many examples of functional differentiation among members of the plant gene family. For example, *FT* promotes flowering in *Arabidopsis*, whereas *TERMINAL FLOWER 1* (*TFL1*), its homolog, inhibits flowering [36]. Likewise, it is known that *AtU2AF65a* promotes flowering, whereas *AtU2AF65b*, its homolog, inhibits flowering in *Arabidopsis* [12,20]. In a previous study, we found that the expression patterns of *FLC* and *FT* genes are major factors in determining the flowering phenotypes of the *atu2af65a* and *atu2af65b* mutants [12]. In the present study, we performed RNA-seq analysis to compare global gene expression patterns of these two mutants. In the case of *atu2af65a* mutants, two mutant alleles were investigated, and approximately 77% of all genes showed abnormal expression in both (Figure 1 and Appendix A). Therefore, we compared the common genes with those abnormally expressed in the *atu2af65b* mutants. Because these are knock-down mutants, in which their remaining functional transcript levels are extremely low compared to their wild-type (Col-0) levels, the differential expression patterns between these mutants are presumably due to their functional differences rather than their mutational strength differences [12].

On examining the expression of flowering time genes registered in FLOR-ID in our RNA-seq data, only a few genes were found to exhibit expression patterns suitable for explaining the flowering time phenotypes of *atu2af65a* and *atu2af65b* mutants (Figure 2, Appendix A). For example, the *atu2af65a* mutants showed increased expression of *FLC* and decreased expression of its downstream target genes, *SOC1* and *AGL24*, whereas there was a decrease in *FLC* expression, but no increase in *SOC1* and *AGL24* expression in the *atu2af65b* mutants (Figure 2). We also found a decrease in *MAF2* and *MAF3* expression, which is also reported in *atu2af65b* mutants by other laboratories [20]. We could not find any change in ABA signaling involved genes such as *ABI5* identified in another study [20]. This discrepancy may be attributable to the use of different *atu2af65b* mutant alleles for this study and seedling samples grown under different growth conditions.

We previously found that FRI complex genes were activated when *FLC* expression increased in the whole seedlings of *atu2af65a* mutants [12]. However, we could not find this kind of concomitant increase in the expression of FRI complex genes in the shoot apices of the *atu2af65a* mutants in this study (Appendix A), suggesting that the increase in *FLC* expression in the *atu2af65a* mutants may be regulated differently in shoot apex regions. Therefore, we examined the expression of other genes that were not found in our initial search for flowering time genes listed in the FLOR-ID (Appendix A). *COOLAIR*, *EDM2*, and *PP2A-b’ɤ* are known to be negative regulators of *FLC* expression [29,34,35,37]. It is known that *Arabidopsis* splicing factors *AtPRP8* (*SUS2*) and *NSRa* are involved in *COOLAIR*-dependent *FLC* expression [30,32]. These two splicing factors play opposite roles in *FLC* gene regulation and only one isoform of each splicing factor functions in *COOLAIR* expression. Similarly, we also found that only the AtU2AF65a protein affected *COOLAIR* regulation (Figure 4 and Figure 5). Changes in *COOLAIR* and *FLC* transcripts were observed in *atu2af65a flc-3* double mutants (Figure 4d). Furthermore, *flc-3* mutation completely suppressed the late-flowering phenotype of *atu2af65a* mutants (Figure 3). These results suggest that *FLC* and the *FLC* upstream regulators are important for the regulation of flowering time in *atu2af65a* mutants.

The coupling of splicing and 3′-end processing of pre-mRNAs is mediated by the interaction between U2AF65 and one of the 3′-end processing factors in humans [38,39]. Similar coupling might also have resulted in the observations in this study (Figure 4). Unlike animal 3′-end processing machinery, plants have a unique RNA-binding protein, FCA, which interacts with the FY 3′-end processing factor to promote poly(A) site selection [40,41]. In *Arabidopsis*, FCA binding of *COOLAIR* transcript is essential for regulating *FLC* expression [42]. It will be interesting to know whether any interaction could occur between AtU2AF65a and 3′-end processing factor such as FCA in *Arabidopsis* as well. Further study is needed to clarify involvement of AtU2AF65a in the AS of *EDM2* and *PP2A-b’ɤ* pre-mRNAs observed in this study.

RNA-seq using chromatin-bound RNA samples is known to identify defects in co-transcriptional splicing in *Arabidopsis* [26,43]. Since we used total RNA samples, we were able to identify a much smaller number of pre-mRNA splicing defects (Figure 6a,b). For example, the relative ratio of four AS transcripts of *AtU2AF65a* pre-mRNA changed in the *atu2af65b* mutant such that isoform 1 increased and isoform 3 decreased (Figure 6e or Figure 7a, Appendix A). The amino acid sequence difference between isoform 1 and isoform 3 protein products is mainly in the UHM domain, which is used for interaction with the AtSF1 protein. In the isoform 1 protein, this UHM domain is damaged, so it is unlikely to function (Appendix A). Therefore, the increase in the functional *AtU2AF65a* isoform 1 in the *atu2af65b* mutants could be due to a compensation mechanism for the defect in AtU2AF65b by increasing AtU2AF65a. The mechanism to compensate for the AtU2AF65b defect by increasing AtU2AF65a through AS regulation could be advantageous over transcriptional regulation because it could be a faster response to stress [44].

AS types in plants are characterized by a higher proportion of intron retention types compared to those in animals. An interpretation of these features was proposed by Jia et al., 2020 [44]. In this proposal, they claim that this “retained intron” can be advantageous in cases where a quick response to stress is required immediately by making a translatable mature mRNA without nascent transcription. This type of regulation at the post-transcriptional splicing level may be applicable for the regulation of AS of *AtU2AF65a* pre-mRNAs, although the switch from isoform 1 to isoform 3 of *AtU2AF65a* AS transcripts was not an intron retention type, but an alternative 3′ splice site type (Figure 6e or Figure 7a, Appendix A). Recently, it was proposed that AtU2AF65a, but not AtU2AF65b, interacts with an RNA-binding protein (CIS) to regulate AS of *FLM* pre-mRNA in ambient temperature-modulated flowering of *Arabidopsis* [45]. Further study on this aspect is required to obtain better insight into the significance of this interaction. 

## 4. Materials and Methods

### 4.1. Plant Material and Growth Conditions

All mutants and wild-type *Arabidopsis* plants used in this study were from the Columbia-0 (Col-0) ecotype background. T-DNA insertion mutants such as *atu2af65a* (SALK_144790 and SALK_075828), *atu2af65b* (SALK_055049c), and *atsf1* (SALK_062177) were obtained from the Arabidopsis Biological Resource Center (ABRC) [12,13]. The *p35S::AtU2AF65* transgenic lines, such as *AtU2AF65a-OX* (*35S:: AtU2AF65a* in *Col-0*) and *AtU2AF65b-OX* (*35S::AtU2AF65b* in *Col-0*), have been described previously [12]. The *atu2af65a-4 flc-3* and *atu2af65b-1 flc-3* double mutants used in this study were generated by crossing between *atu2af65a-4* or *atu2af65b-1* mutants with *flc-3* mutants. Seeds were surface-sterilized three times with 70% (*v/v*) ethanol added 0.5% (*v/v*) triton X-100, and two times with 95% (*v/v*) ethanol. Sterilized seeds were stratified at 4 °C for 3 d prior to germination. Plants were grown on 1/2 strength Murashige and Skoog (MS) medium or in soil (Sunshine Mix 5, Sun Gro Horticulture, Agawam, MA, USA) at 23, 16, or 10 °C under long-day (16-h light/8-h dark) conditions at a light intensity of 120 μmol m^−2^ s^−1^.

### 4.2. RNA Extraction, RT-PCR and Real-Time Quantitative PCR 

Total RNA was extracted from 7 d-old whole seedlings using the TRIzol Reagent (Invitrogen, Carlsbad, CA, USA). RNA was quantified using a Nanodrop ND-2000 spectrophotometer (Thermo Scientific, Waltham, MA, USA) and its integrity was checked. Complementary DNA (cDNA) was synthesized using M-MLV Reverse Transcriptase (Invitrogen) with 1 μg of RNA and 1 μg of oligo(dT_15_) primers according to the manufacturer’s instructions. The reaction was performed at 42 °C for 90 min, with preincubation at 72 °C for 5 min and enzyme inactivation at 72 °C for 15 min. For semi-quantitative reverse transcription polymerase chain reaction (RT-PCR) analysis, cDNA was amplified with *Taq* polymerase using gene-specific primers under the following thermal cycling conditions: pre-incubation at 95 °C for 5 min, 22 cycles of 95 °C for 30 s, 59 °C for 30 s, and 72 °C for 30 s; and a final extension at 72 °C for 5 min. Real-Time Quantitative PCR (RT-qPCR) was performed in 384-well plates of a LightCycler 480 using a LightCycler 480 SYBR Green I Master mixture (Roche Applied Science, Madison, WI, USA). All semi-quantitative RT-PCR or RT-qPCR experiments were conducted in three biological replicates (independently harvested samples), with three technical replicates each. The primers used in this study are listed in Appendix A.

### 4.3. Preparation of Shoot Apices and RNA-seq

Shoot apices were sampled from stem-root cuttings of seven-day-old whole seedlings using micro vannas scissors (Appendix A). RNA-seq libraries were constructed for 12 RNA samples according to the instructions for TruSeq RNA library preparation Illumina (https://www.illumina.com, accessed on 10 April 2023, 1000000040498 v00). In these preparations, poly(A)-containing mRNA molecules were purified using poly T oligo-attached magnetic beads, RNA was fragmented at elevated temperatures, and random hexamers were used for first-strand cDNA synthesis. The 48 libraries had an average insert size of 120–200 bp with a median size of 150 bp, and each library was sequenced on Illumina sequencing platforms to generate paired-end reads. The total number of mapped reads generated in the RNA-seq data was approximately 53 million paired-end reads per sample (Appendix A). Read coverage was calculated as the read number per million total sequencing reads per base pair. Genes with a log2 fold change (log2 FC) were considered to be DEGs between each sample and the control (*q*-value < 0.05). Genes with multiple transcripts were considered alternatively spliced when they exhibited a transcript change in more than one of the transcript DEGs (*p*-value < 0.05). Variations in splice isoform expression were examined by comparing differential expression at gene and transcript levels. Differential AS events such as major categories (i.e., SE, A5SS, A3SS, MXE, and RI) in RNA-seq data (FDR < 0.05) were detected using rMATS. Each mapped splice junction read required at least 8 bp (anchor length) from each side of the splice junction [18], and all AS events, including duplicate reads, were identified based on rMATS (Appendix A). Considering the association between transcript expression and AS events for partial reads, two or more transcripts and rMATS were compared (Appendix A). GO analysis was performed online using ShinyGO 0.76.3 (http://bioinformatics.sdstate.edu/go/ accessed on 10 April 2023) [46]. These data have been submitted to Experiment ArrayExpress (https://www.ebi.ac.uk/fg/annotare/edit/7612/ accessed on 10 April 2023) (accession: E-MTAB-12648).

### 4.4. Nuclear/Cytoplasmic Fractionation and Isolation of Nuclear Extracts

Nuclear extraction from seven-day-old *Arabidopsis* seedlings was performed using a previously proposed method [26]. Approximately 0.5 g of plant tissues per 2 mL tube was ground with liquid nitrogen in the tube and gently mixed with 800 μL of Honda buffer (0.5% (*v*/*v*) Triton X-100, 1× protease inhibitor mixture (Roche), 20 U/mL RNase inhibitor (Thermo Scientific), 50 ng/μL yeast tRNA (Invitrogen) and syringe filtered mixture of 20 mM Hepes, 0.44 M sucrose, 1.25% (*w*/*v*) Ficoll, 2.5% (*w*/*v*) Dextran T40, 5 mM dithiothreitol (DTT), 10 mM MgCl_2_). Tissue homogenates were filtered through two layers of Miracloth and centrifuged at 11,000× *g* for 10 min. The supernatant was used as cytoplasmic fraction, and pellet was resuspended in 1 mL of resuspension buffer (50% (*v*/*v*) glycerol, 0.5 mM EDTA, 25 mM Tris-HCl (pH 7.5), 100 mM NaCl, 1 mM DTT, 0.4 U/mL RNasin RNase inhibitor (Promega, WI, USA), 100 ng/mL yeast tRNA). The nuclear suspension was resuspended by adding 400 μL of urea buffer (1 M urea, 0.5 mM EDTA, 25 mM Tris-HCl (pH 7.5), 300 mM NaCl, 1 mM DTT, 1% (*v*/*v*) Tween 20) and then centrifuged at 2000× *g* for 15 min. The supernatant was removed carefully and washing with urea buffer was repeated. The cytoplasmic nuclear fractions were vortexed in 600 μL TRIzol reagent for 30 s. The extracted RNA was quantified using a Nanodrop spectrophotometer and checked for integrity. Later, when RNA was used, *miR167a*, which enriched in the nuclear fraction compared to the total fraction [47], was used as the nuclear extract control. For very small quantities of samples, similar results were obtained by RNA extraction using the Cytoplasmic and Nuclear RNA Purification Kit (Norgenbiotek, Thorold, ON, Canada).

### 4.5. Statistical Analysis

To reveal statistical differences in RNA expression experiments, one-way analysis of variance (ANOVA) was performed using the Duncan method with SPSS software (version 25, IBM, Armonk, NY, USA) or two-tailed paired Student’s *t*-tests of variances in Microsoft Excel.

## 5. Conclusions

In this study, we performed RNA sequencing (RNA-seq) of shoot apices from *atu2af65a* and *atu2af65b* mutants and compared their transcriptomes. Among the genes with opposite expression patterns in the two mutants, *FLC* and a *non-coding RNA* (*At4g04223*) were the most significant. We also observed changes in the expression or alternative splicing (AS) of some *FLC* upstream regulators, such as *COOLAIR*, EDM2, or PP2A-b’ɤ, in the shoot apices of these mutants. Moreover, we discovered that *atu2af65b* mutation, *atsf1* mutation, and low temperature conditions increased the levels of a functional *AtU2AF65a* isoform, suggesting a possible compensation mechanism involving the AS of *AtU2AF65a* pre-mRNA. Our results indicate that AtU2AF65a and AtU2AF65b splicing factors modulate *FLC* expression by affecting the expression or AS of a subset of *FLC* upstream regulators in the shoot apex, leading to opposite flowering phenotypes. 

## Figures and Tables

**Figure 1 plants-12-01655-f001:**
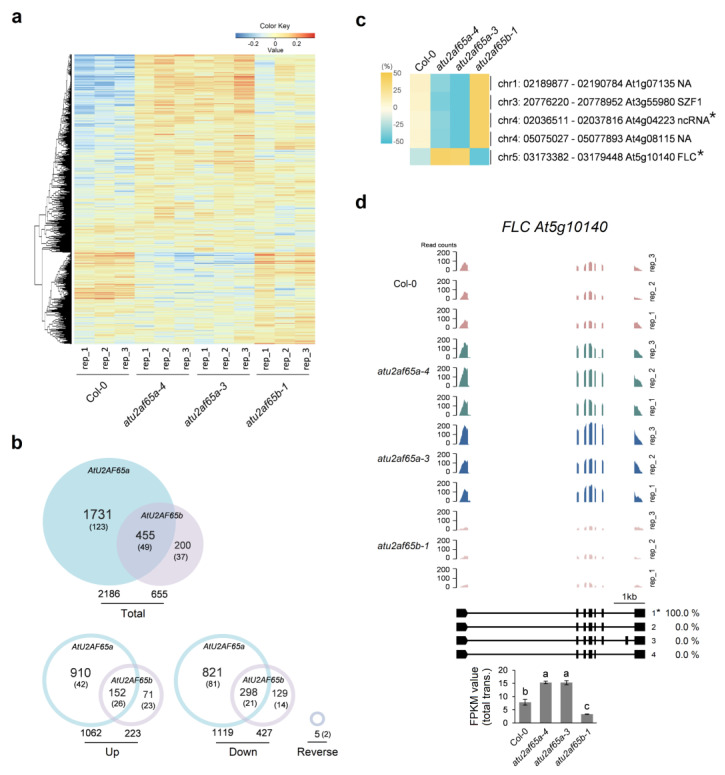
Identification and analysis of differentially expressed genes (DEGs) in shoot apices of wild-type (Col-0), *atu2af65a-4*, *atu2af65a-3*, and *atu2af65b-1* plants. (**a**) Heatmap showing the hierarchical clustering for the genes expressed in Col-0, two *atu2af65a*, and one *atu2af65b* plants. (**b**) Venn diagrams showing the total number of DEGs regulated by *AtU2AF65a* and *AtU2AF65b* (top), and the number of up, down, and reverse DEGs (bottom). Reverse indicates opposite DEGs between *atu2af65a* and *atu2af65b* mutants. Numbers in parentheses indicate the genes (|log_2_ FC| ≥ 1). *q*-value (*q* < 0.05) provides a means to control the positive false discovery rate. (**c**) Five genes that were reversely regulated in *atu2af65a* and *atu2af65b* mutants. Asterisks indicate the genes (|log_2_ FC| ≥ 1). (**d**) Coverage plot and gene expression (measured as fragments per kilobase of transcript per million mapped reads [FPKM]) showing reads mapping to the *FLC* locus in shoot apices of Col-0, *atu2af65a-4*, *atu2af65a-3*, and *atu2af65b-1* plants. Wiggle plots were generated for three biological replicates for sequencing data in the integrative genomics viewer (IGV). The letters shown in FPKM value graph indicate statistically significant differences (*p* < 0.05) based on the Duncan method. The gene structures below the wiggle plots represent four different *FLC* isoforms. One major *FLC* isoform (1) is indicated with an asterisk and its FPKM value is 100%.

**Figure 2 plants-12-01655-f002:**
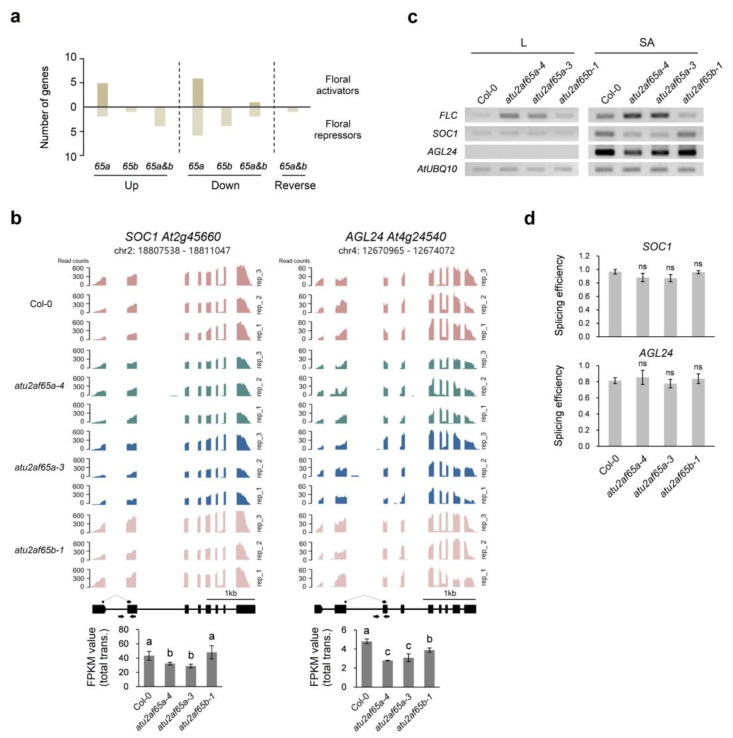
*AtU2AF65a* positively regulates *SOC1* and *AGL24* via *FLC* in shoot apex regions. (**a**) Comparison between 306 flowering time genes in FLOR-ID and differentially expressed genes (DEGs) from our RNA sequencing analysis. *65a*, DEGs of *atu2af65a* mutants only; *65b*, DEGs of *atu2af65b* mutants only; *65a*&*b*, overlapped DEGs of *atu2af65a* and *atu2af65b* mutants. (**b**) Coverage plot and gene expression (measured as FPKM) showing reads mapping to the *SOC1* and *AGL24* loci in shoot apices of wild-type (Col-0), *atu2af65a-4*, *atu2af65a-3*, and *atu2af65b-1* plants. Wiggle plots were generated for three biological replicates for sequencing data in the IGV. The letters indicate statistically significant differences (*p* < 0.05) based on Duncan method. (**c**) Expression of *FLC*, *SOC1*, and *AGL24* in leaves (L) and shoot apices (SA) of Col-0, *atu2af65a-4*, *atu2af65a-3*, and *atu2af65b-1* plants using semi-quantitative RT-PCR analysis. *AtUBQ10* (*At4g05320*) gene was used as an internal control. (**d**) Splicing efficiency of *SOC1* or *AGL24* in nuclear RNA fraction of shoot apex regions measured by RT-qPCR analysis. Splicing efficiency was calculated according to Ding and Elowitz [25]. The primer positions are indicated in black arrows shown in (**b**). Student’s *t*-test was used for statistical analysis. ns, not significant.

**Figure 3 plants-12-01655-f003:**
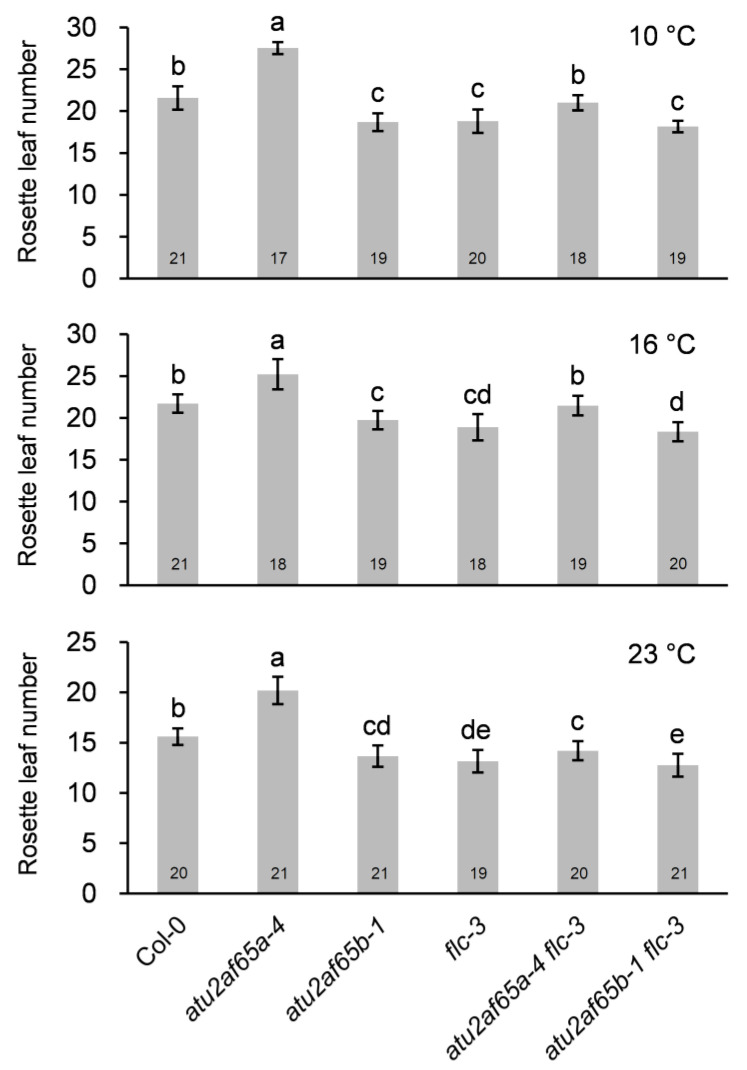
Flowering time of wild-type (Col-0), *atu2af65a-4*, *atu2af65b-1*, *flc-3*, *atu2af65a-4 flc-3*, and *atu2af65b-1 flc-3* plants at different temperatures under long-day (LD) conditions. The letters indicate statistically significant differences (*p* < 0.05) based on the Duncan method. Numbers in the bars represent the plants measured.

**Figure 4 plants-12-01655-f004:**
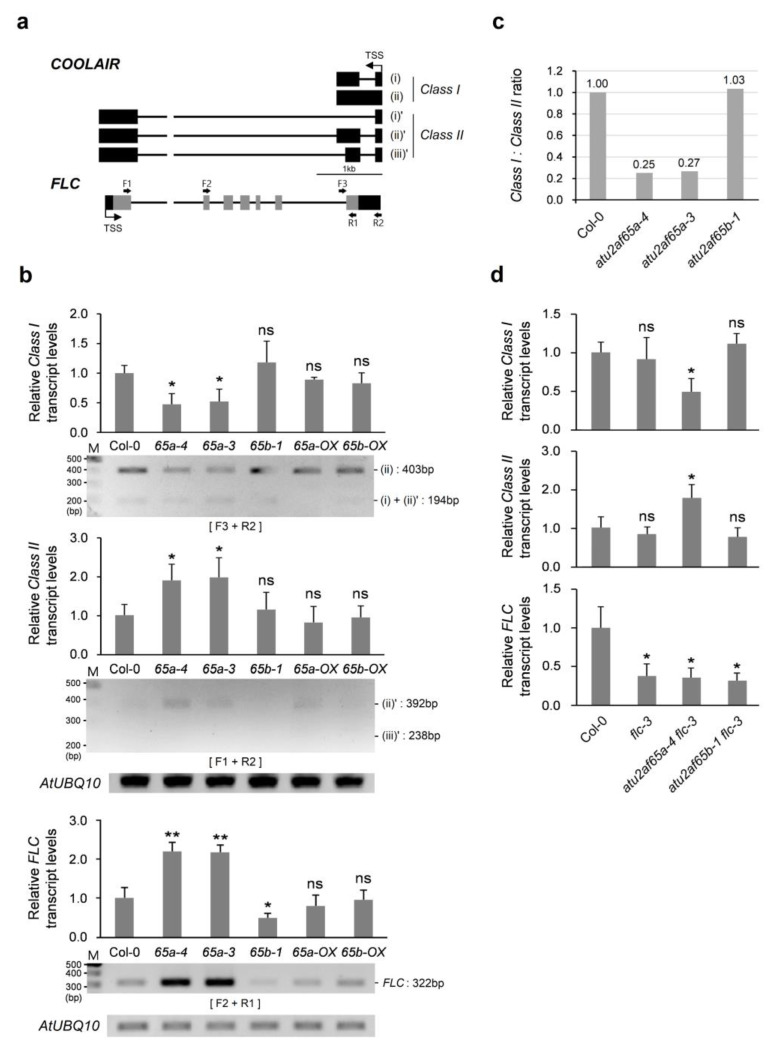
Alteration in *AtU2AF65a* affects *COOLAIR* transcripts in shoot apices. (**a**) Schematic representation for *COOLAIR* and *FLC* at the *FLC* genomic region. The grey and black rectangles indicate coding and noncoding regions, respectively. TSS, transcription start site; F1 to 3, forward primers; R1 to 2, reverse primers. (**b**) Expression of *COOLAIR Class I*, *COOLAIR Class II*, and *FLC* in shoot apices of wild-type (Col-0), *atu2af65a-4*, *atu2af65a-3*, *atu2af65b-1*, *AtU2AF65a-OX*, and *AtU2AF65b-OX* plants. *IPP2* (*At3g02780*) and *AtUBQ10* (*At4g05320*) genes were used as internal controls for RT-qPCR and semi-quantitative RT-PCR analyses, respectively. The primers used for semi-quantitative RT-PCR analysis denoted by (**a**). (**c**) The ratio of *COOLAIR Class I* to *COOLAIR Class II* isoforms of *COOLAIR* transcripts relative to Col-0 from RT-qPCR data. (**d**) Expression of *COOLAIR Class I*, *COOLAIR Class II*, and *FLC* in shoot apices of each genotype as measured by RT-qPCR analysis. Asterisks indicate statistically significant differences (* *p* < 0.05, ** *p* < 0.01) based on Student’s *t*-test. ns, not significant.

**Figure 5 plants-12-01655-f005:**
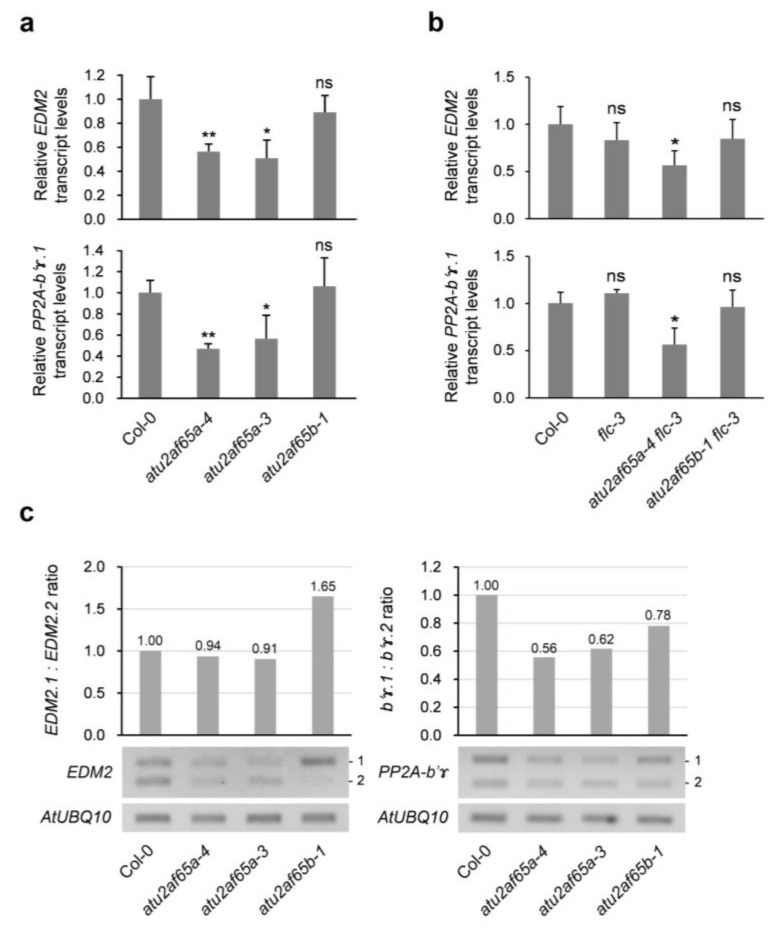
AtU2AF65a or AtU2AF65b control the expression or alternatively spliced transcripts of *EDM2* or *PP2A-b’ɤ* in shoot apex regions. (**a**) Expression of *EDM2* and *PP2A-b’ɤ* in shoot apices of wild-type (Col-0), *atu2af65a-4*, *atu2af65a-3*, and *atu2af65b-1* plants. *IPP2* (*At3g02780*) gene was used as an internal control for RT-qPCR analysis. (**b**) Expression of *EDM2* and *PP2A-b’ɤ* in shoot apices of *atu2af65a-4 flc-3* and *atu2af65b-1 flc-3* double mutants using RT-qPCR analysis. (**c**) The ratio of *EDM2.1* to *EDM2.2* isoforms of *EDM2* and *PP2A-b’ɤ.1* to *PP2A-b’ɤ.2* isoforms of *PP2A-b’ɤ* transcripts relative to Col-0 as based on FPKM values from RNA sequencing data (top) and measured by semi-quantitative RT-PCR analysis (bottom). Asterisks indicate statistically significant differences (* *p* < 0.05, ** *p* < 0.01) based on Student’s *t*-test. ns, not significant.

**Figure 6 plants-12-01655-f006:**
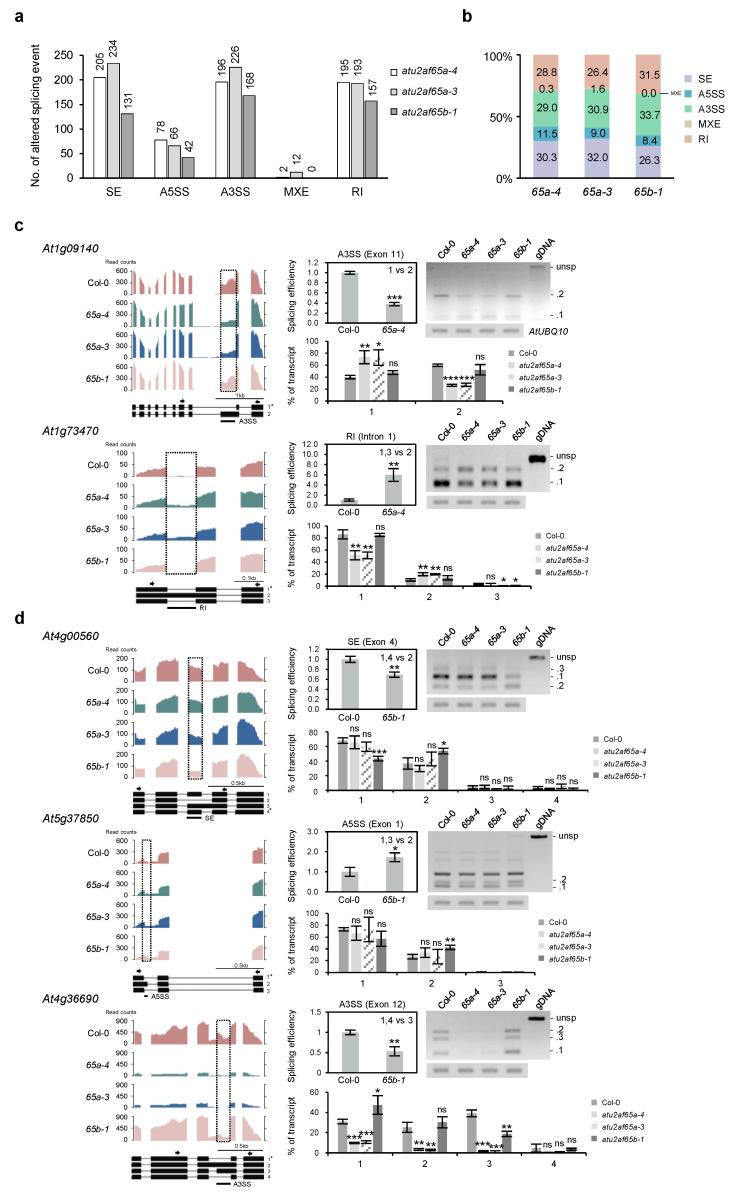
AtU2AF65a and AtU2AF65b are required for alternative splicing of pre-mRNAs in shoot apex regions. (**a**,**b**) The numbers and proportion of altered splicing events in shoot apices of *atu2af65a-4*, *atu2af65a-3*, and *atu2af65b-1* mutants compared to wild-type (Col-0) plants. SE, skipped exon; A5SS, alternative 5′ splice site; A3SS, alternative 3′ splice site; MXE, mutually exclusive exons; RI, retained intron. (**c**) Coverage plot showing the reads mapping to the *At1g09140* and *At1g73470* loci selected from Appendix A and their splicing efficiencies and gene expression in shoot apices of each genotype. (**d**) Coverage plot showing the reads mapping to the *At4g00560*, *At5g37850*, and *At4g36690* loci selected from Appendix A and their splicing efficiencies and gene expression in shoot apices of each genotype. unsp, unspliced. Wiggle plots were generated for one representative out of the three biological replicates for sequencing data in the IGV. Splicing efficiencies were based on rMAT analysis of RNA sequencing, and expression was measured by semi-quantitative RT-PCR analysis. Percentage of transcripts was converted based on one hundred percent of total transcripts in FPKM value of Col-0. Asterisks indicate statistically significant differences (* *p* < 0.05, ** *p* < 0.01, *** *p* < 0.001) based on Student’s *t*-test. ns, not significant. unsp, unsplicied transcript.

**Figure 7 plants-12-01655-f007:**
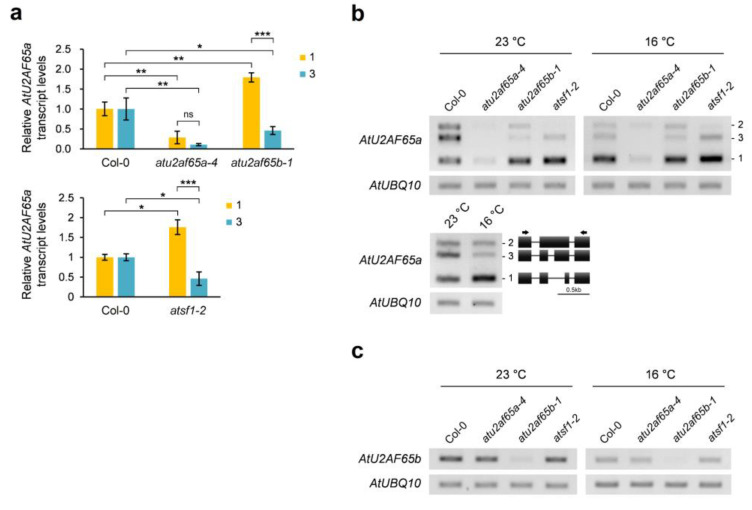
Changes of alternative splicing variants of *AtU2AF65a* and *AtU2AF65b* in *atu2af65b* and *atsf1* mutants at different ambient temperatures. (**a**) Expression of alternatively spliced *AtU2AF65a* mRNA forms in seven-day-old whole seedlings of wild-type (Col-0), *atu2af65a-4*, *atu2af65b-1*, and *atsf1-2* plants at 23 °C. *IPP2* (*At3g02780*) gene was used as an internal control for RT-qPCR analysis. Asterisks indicate statistically significant differences (* *p* < 0.05, ** *p* < 0.01, *** *p* < 0.001) based on Student’s *t*-test. ns, not significant. (**b**) Expression of alternatively spliced *AtU2AF65a* mRNA forms in seven-day-old whole seedlings of Col-0, *atu2af65a-4*, *atu2af65b-1*, and *atsf1-2* at 23 °C and in ten-day-old whole seedlings at 16 °C. The primer positions are indicated by black arrows. (**c**) Expression of *AtU2AF65b* in whole seedlings of Col-0, *atu2af65a-4,* and *atu2af65b-1* plants at 23 °C and 16 °C. *AtUBQ10* (*At4g05320*) gene was used as an internal control for semi-quantitative RT-PCR analysis.

## Data Availability

RNA-seq data was deposited in E-MTAB-12648.

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
