# Peer review of "Two *Arabidopsis* Splicing Factors, U2AF65a and U2AF65b, Differentially Control Flowering Time by Modulating the Expression or Alternative Splicing of a Subset of *FLC* Upstream Regulators"

_plants, 2023, doi:10.3390/plants12081655_

Round 1

Reviewer 1 Report

In current study, Lee et al., reported the RNA-seq based analysis of shoot apical samples of atu2af65a and atu2af65b mutants, the mutant shared only 49 out of 209 genes with a two-fold or higher change in expression levels. Further analysis revealed FLOWERING LOCUS C (FLC), as a major flowering repressor gene and involvement of a non-coding RNA (At4g04223). The role of ncRNA is not studied well. Moreover, the authors noticed alternative splicing (AS) patterns of COOLAIR, EDM2, or PP2A-b’ɤ, the FLC upstream regulators were unique.

Few comments:

In abstract provide most significant results/outcomes observed in this study.

I suggest authors to enrich some background information on alternative splicing. In addition to RNA-seq, there are other methods such as Raman sensing (https://doi.org/10.1111/tpj.13537 and https://doi.org/10.1016/j.febslet.2014.02.061).

Figure 1. The figure panels could be enlarged to see clearly. The labels are difficult to visualize.

Authors may consider to include a conclusions section.

Overall, the study indicated AtU2AF65a and AtU2AF65b isoforms regulate FLC expression and affect the plant flowering phenotypes reciprocally.

The study is methodologically conducted and carries high significance in field of alternative splicing regulation in plants.

Reviewer 2 Report

The authors investigated a set of splicing factors, U2AF65-a and -b. Since the splicing factors can generate several transcripts from a single transcript and thus display a variety of phenotypes, the understandings of the effect on transcriptomes are very important. They found that opposing effect of these transcription factors on FLC expression by RNA-seq analysis. Moreover, these transcription factors can modulate the gene expression networks upstream of FLC, and results in flowering sensitivity. The study is performed straight-forward, and the methods used are appropriate. Although the gene-network studies tend to lead to ambiguous conclusions, the author’s direct methodology is considered to be effective.

Author Response

We appreciate the comment.

Reviewer 3 Report

The authors describe the mechanism explaining how a subset of spliceosome factors differentially regulate FLC upstream regulators. The work is of high interest to the scientific community as it links the activity of the spiceosome with temperature.

I believe that the introduction needs to be improved, the sentences rearranged to significantly increase readability and quality (e.g. lines 32-34; 43-44; 48-49; 53; 58-61). Furthermore, it needs to be clear in the introduction in which organs FLC is misrelated in the mutants. This will help the authors justify why SAM samples were chosen to perform the RNA-seq. It needs to be clear why the authors did not choose leaves.

The results section also requires improvement. Why is it important to use 2 65a mutant lines and not 2 65b lines? What is the difference between the lines?

Is it not more robust to immediately exclude all the transcripts below two fold change (Lines 106-108)?

I do not agree with the authors in the conclusion they used on lines 122-125. This may be because the previous paragraph is quite dense and needs restructuring.

The introduction to the need to study temperature and the double mutants should be better introduced in the text (lines 198-202). And although the 65b mutation is not additive to flc-3 I do not agree that the flc-3 completely supresses the 65a phenotype (Figure 3) which might indicate additional factors independent of flc influencing flowering time in the mutant. I would avoid the use of the words completely when describing the interaction between the genes.

How does alternative splicing of EDM2 and PP2A could translate into changes in flc activity? Maybe explore the function of the alternative transcripts (Lines 289-290) in 65b. Is there any change at the protein level?

The figures are in general small, particularly figure 6 and thus very difficult to understand.

It would be interesting to have a final figure with the mechanism explained. 
